# Nutritional Composition of Honey Bee Drones of Two Subspecies Relative to Their Pupal Developmental Stages

**DOI:** 10.3390/insects12080759

**Published:** 2021-08-23

**Authors:** Sampat Ghosh, Pascal Herren, Victor Benno Meyer-Rochow, Chuleui Jung

**Affiliations:** 1Agriculture Science and Technology Research Institute, Andong National University, Andong 36729, Korea; sampatghosh.bee@gmail.com (S.G.); meyrow@gmail.com (V.B.M.-R.); 2Institute of Natural Resource Sciences, Zürich Campus Grueental, University of Applied Sciences (ZHAW), 8820 Waedenswil, Switzerland; pahe@plen.ku.dk; 3Department of Genetics and Ecology, Oulu University, 90140 Oulu, Finland; 4Department of Plant Medicals, Andong National University, Andong 36729, Korea

**Keywords:** *Apis mellifera carnica*, *Apis m. mellifera*, beekeeping, health, amino acids, fatty acids, minerals, bee hive products, alternative food

## Abstract

**Simple Summary:**

Despite the use of honey bee brood as food among several communities of the world, the nutritional potential of drones remained unexplored for a long time. In the recent past some scientific endeavour, including our own previous work, has been undertaken to explore the nutrient quality of this food source. Due to their limited socio-biological role, honey bee drones would be a suitable candidate to compare their nutrient content with that of worker honey bees. We therefore investigated the nutrient composition of honey bee drones belonging to two subspecies, namely *Apis mellifera carnica* and *A. m. mellifera* covering their pupal developmental period. To possess information of the drones’ nutritional value during their development would help in choosing the most suitable developmental stage for the commercial production of drone brood as food.

**Abstract:**

We examined the contents of nutritional importance, i.e., amino acids, fatty acids and minerals of different developmental stages of drones of two honey bee subspecies, namely *Apis mellifera carnica* and *A. m. mellifera*. The results revealed that, in general, individual amino acid amounts and therefore the total protein increased along with the developmental stages of the drones. No statistically significant differences were found between the same developmental stages of the two subspecies. The reverse, i.e., a decrease with developmental stage occurred in relation to the fatty acid composition. Most of the minerals were higher at advanced developmental stages. Overall, the high protein content (31.4–43.4%), small amount of fat (9.5–11.5%) and abundance of minerals such asiron and zinc, make drones a suitable nutritional resource. Even though nutrient content, especially protein, was higher in the pupae than the prepupae, we propose prepupae also as a commercial product based on their higher biomass production. Provided standard production protocols maintaining hygiene and safety will be adhered to, we propose that drone honey bees can be utilized as human food or animal feed.

## 1. Introduction

Humans have interacted with honey bees since ancient times. Archaeological evidence such as Egyptian bee iconography dating back to approx. 2400 BC, beeswax found in the pottery from Europe and Africa [1,2], the existence of prehistoric cave paintings depicting honey collection [3] and the discovery of an apiary of an industrial nature from the Iron Age at Tel Reḥov during the Beth-Shean Valley archaeological project by [4] all testify to this age-old connection between humans and bees.

Keeping honey bees and looking after them is known from the antiquity and appears to have arisen in various parts of the world with a focus on Asia. The primary purpose of maintaining honey bee colonies was the production of honey and beeswax. Honey is mentioned in the age-old literatures of the Ayurveda, the Talmud, the Bible and the Quran and has acquired a reputation as an anti-microbial, anti-oxidant, anti-ageing substance, potent in dealing with numerous disorders [5,6,7].

Nowadays honey bees are not just seen as honey and wax producers, but also a source of bee pollen, royal jelly, propolis and pharmaceutically appreciated bee venom. However, despite the practice of consuming bee brood in several parts of the world (to name but a few places: Australia: [8]; Zambia: [9]; Tanzania: [10]; Thailand: [11]; Mexico: [12]; Ecuador: [13]), honey bee brood itself has received comparatively little appreciation.

Information on the nutritional value of bee brood, in contrast to studies dealing with other edible insects [14,15,16,17,18,19,20,21,22,23,24], is scarce and limited to only a few publications. To cite some examples: there are reports on the nutrient content of honey bee brood (mostly pupae and 10% larvae, although species and subspecies have not been mentioned) by Finke [25], on *Apis mellifera* adults by Banjo et al. [15], on *Apis mellifera ligustica* worker bees from Korea at different stages of development by Ghosh et al. [26] and on *A. dorsata* and *A. cerana* workers from Thailand at different developmental stages by Ghosh et al. [27]. However, drone brood is more suitable as food for human consumption than worker bee brood, because of the drones’ limited socio-biological role in the hive. Moreover, drones are larger than worker bees in size and they are often removed from the colony to control parasites such as the destructive *Varroa* mite.

Given these factors, our objective for the present study was to focus on honey bee drones and to specifically examine their nutrient composition at different developmental stages. We decided to compare the nutrient composition of drones and their growth stages of two different honey bee subspecies, namely *Apis mellifera carnica* and *A. m. mellifera* obtained from the same ecological environment. This was accomplished because it had been shown for the pollen collecting bee *Osmia bicornis* that the sexes differed fundamentally in the assimilation and allocation of acquired atoms, elemental phenotypes and stoichiometric niches for components such as “food (pollen), eggs, pupae, adults, cocoons and excreta” [28] and that published data available for worker bees might not be applicable to drones. Moreover, since the production of drone honey bees does not require any high throughput technology, an inclusion of drones and finding a use for them as another hive product could help enhancing the economy of small to medium scale bee farmers.

## 2. Materials and Methods

### 2.1. Sample Collection

*Apis mellifera carnica* and *A. m. mellifera* drone samples were collected from private apiaries located in Wölflinswil and Zürich, Switzerland, respectively, during the month of May in 2019. Three combs of capped brood were harvested from three different healthy bee colonies (in both locations). Each comb was individually vacuum-packed, sealed and immediately kept in freezing box at −20 °C. The combs were transported in their frozen state to the laboratory and stored in the refrigerator (F500, Hettich, Bäch SZ, Switzerland) at −20 °C. Prepupae, white-eyed pupae, dark-eyed pupae and adults were separated from the wax by breaking the frozen combs and collecting the insects with tweezers [29]. The samples were freeze dried for at least 72 h at −55 °C using the laboratory freeze dryer (ALPHA 1-2 LDplus, Christ, Osterode am Harz, Germany). Body weights on the basis of dry matter of different developmental stages (n = 20 for every developmental stage) of both subspecies were recorded using Acculab laboratory weighing balance (ALC310.3, Kern, Kingswinford, West Midlands, UK). The samples (n = 20 for every developmental stage) were ground to a powder for further analysis.

### 2.2. Identification of the Species

The specimens, i.e., two subspecies of *Apis mellifera* namely *Apis mellifera carnica* and *Apis mellifera mellifera,* were identified based on DNA barcoding to confirm the molecular level identification. The total DNA of each sample was extracted from the front part in case of prepupae and thorax in case of pupae using a DNeasy Blood and Tissue kit (QIAGEN, Inc., Dűsseldorf, Germany) following the protocol supplied by the manufacturer. Two primers, LCO-1490 (50-GGT CAA CAA ATC ATA AAG ATA TTG G-30) and HCO-2198 (50-TAA ACT TCA GGG TGA CCA AAA AAT CA-30) targeting the mitochondrial Cytochrome Oxidase I (COI) gene [30] were used. The polymerase chain reaction (PCR) was carried out in order to amplify the cytochrome oxidase subunit 1 (COI) gene corresponding to “DNA Barcode” region [31] using AccuPower PCR PreMix (Bioneer, Daejeon, Korea). Sequencing was carried out by Macrogen Inc. (Seoul, Korea) commercially. The sequences (obtained in both directions) were assembled with the help of Bioedit v7.0.5.2 program. We BLAST (Basic Local Alignment Search Tool) the sequences to the database of the National Center for Biotechnology Information (NCBI) (http://www.ncbi.nlm.nih.gov, accessed on 1 June 2021) to confirm the specimen up to subspecies level with the available in the database.

### 2.3. Amino Acid Analysis

The amino acid composition was estimated using a Sykam Amino Acid analyzer S433 (Sykam GmbH, Germany) equipped with Sykam LCA L-07 column following the standard method given in [32]. 20 mg of each sample was hydrolyzed in 6 N HCl for 24 h at 110 °C under a nitrogen atmosphere and then concentrated by the rotary evaporator (EYELA N1001, Tokyo, Japan). The concentrated samples were reconstituted with sample dilution buffer supplied by the manufacturer (physiological buffer 0.12 N citrate buffer, pH 2.20) and analyzed for amino acid composition.

### 2.4. Fatty Acid Analysis

Fatty acids were estimated following the standard method of the Korean Food Standard Codex [33] by gas chromatography-flame ionization detection (GC-14 B, Shimadzu, Tokyo, Japan) equipped with anSP-2560 column. The samples were derivatized into fatty acid methyl esters (FAMEs). Identification and quantification of FAMEs were accomplished by comparing the retention times of peaks with those of pure standards purchased from Sigma (Yongin, Korea) and analyzed under the same conditions.

### 2.5. Mineral Analysis

Minerals of nutritional importance namely calcium, magnesium, sodium, potassium, phosphorus, iron, zinc and copper were analyzed following standard procedures according to the Korean Food Standard Codex [33]. Each sample was digested with nitric and hydrochloric acid (1:3) at 200 °C for 30 min., filtered using Whatman filter paper (0.45 micron) and stored in washed glass vials before analyses could commence. The mineral contents were analyzed using an inductively-coupled plasma-optical emission spectrophotometer (ICP-OES 720 series; Agilent; Santa Clara, CA, USA).

### 2.6. Statistical Analysis

Composite sampling methods were followed for each group. In order to increase reliability all the chemical analyses were carried out in triplicate and represented as mean ± standard deviation. To examine the differences in body weight as well as individual nutrients in connection with the developmental stages, we carried out one way ANOVA (analysis of variance) and post hoc tests (Tukey HSD) using SPSS 16.0. *t*-tests were carried out to examine the differences of individual nutrients at identical developmental stages of the two honey bee subspecies. If the *p* value was found ≤0.05 (CI = 95%), the null hypothesis was rejected.

## 3. Results

### 3.1. Identification of Specimens

With the help of the DNA barcoding method we confirmed the specimens as *Apis mellifera carnica* and *Apis mellifera mellifera*. The alignment report has been presented in the Appendix A.

### 3.2. Body Weights and Sizes

Figure 1 represents the comparative body weight changes (on dry matter basis) along with the developmental stages of the drones. The results revealed that the body weights in both subspecies decreased significantly as the development progressed (for *A. m. carnica*: df = 3, 76, F = 328.039, *p* = 0.000; *A. m. mellifera*: df = 2, 57, F = 89.652, *p* = 0.000). Regarding comparisons between the two subspecies, the weights of dried *A. mellifera carnica* drone prepupae (103.3 mg) and white eyed pupae (90.3 mg) were significantly higher than those of *A. mellifera mellifera* (prepupae: 96.6 mg; white eyed pupae: 84.4 mg). However, no significant difference was found in connection with dried body weights in dark eyed pupae of the two subspecies. A similar trend was observed in body length (dried). The average length of an *A. mellifera carnica* prepupa (15.6 mm) was significantly higher than that of an *A. mellifera mellifera* prepupa (14.8 mm). The same trend was found in dark eyed pupae (*A. m. carnica*: 15.7 mm and *A. m. mellifera*: 15.3 mm). However, irrespective of the developmental stage no difference was apparent in case of body width.

### 3.3. Amino Acid Composition

Table 1 represents the amino acid composition of the different developmental stages of *A. m. carnica* and *A. m. mellifera* drones. There is a statistically significant difference (df = 3.4; *p* ≤ 0.05; CI = 95%) in the amino acid contents of the different developmental stages. However, in most cases no statistically significant differences were found in relation to individual amino acids, with the exception of a few (such as glycine and glutamic acid for dark eyed pupae), when drone brood of the same developmental stage of both subspecies was examined (Table 1). Irrespective of subspecies and developmental stages, glutamic acid was the most abundant amino acid overall, but among the essential amino acids leucine was predominating followed by lysine. Figure 2 represents the scoring pattern of the essential amino acids of the different developmental stages of the drones, compared with the amended value of WHO/FAO/UNU 2007 report [34].

### 3.4. Fatty Acid Composition

Table 2 represents the fatty acid profiles of different developmental stages of *A. m. carnica* and *A. m. mellifera* drones. No statistically significant differences in the total fatty acid contents were found between prepupae and white eyed pupae. However, total fatty acid content significantly decreased at the later stage, dark eye pupae in both subspecies. Polyunsaturated fatty acids (PUFA) were the least abundant and saturated fatty acids (SFA) were predominating followed by monounsaturated fatty acids (MUFA) in all the cases. However, in contrast with the situation of the amino acids, individual fatty acid content differed significantly between the same developmental stages of the two subspecies. Palmitic acid followed by stearic and myristic acids were dominated the SFAs. Oleic acid was the most abundant among the MUFAs.

### 3.5. Mineral Content

Table 3 represents the mineral contents of the two subspecies. The increasing trend of individual mineral content with developmental stage is noteworthy. Potassium was the most abundant mineral available in drone brood, but sodium content was the least abundant of the macro minerals. Table 4 represents the satisfaction level of RDA (Recommended Dietary Allowance) for different minerals of nutritional importance by the consumption of 100 g of drones of different developmental stages. Figure 3a–f represents the comparative account of mineral contents with that of conventional meats i.e., chicken, beef and pork. In general, drones irrespective of developmental stage, were found to contain a higher amount of minerals, except for sodium, than that of conventional meats, i.e., beef, chicken, pork.

## 4. Discussion

As pointed out in numerous earlier publications, to name but a few, [23,37,38,39,40,41,42,43] ever since it was first suggested by Meyer-Rochow in 1975 [44], the use of insects as food for humans and feed for animals has until quite recently been given rather little attention. However, the advantages of making greater use of insects are obvious when seen in the context of health-related and ecological issues and with regard to global climate change [45,46,47,48,49,50]. At least 2000 different species of insects are accepted as food worldwide [51] and increasingly more species are having their chemical compositions analysed. However, honey bees and their drones, as stated already in the Introduction, feature hardly at all in the analyses of nutrient contents and chemical compositions of insects.

Honey bees are also rarely included in studies on the acceptability of insects as human food [52,53,54] or comparisons with more conventional human food sources [21,55,56,57]. The emphasis, with few exceptions such as the study by Nyberg et al. [58] has been to find reasons for the widespread neophobia that apparently prevents especially, but not exclusively [59], people of western cultural backgrounds to accept insects as food. In spite of the difficulties to popularise dishes and food items based on or containing insects and their products, the provocative question “Insects as food—is the future already here?” was asked [60] and cautiously answered in the affirmative. Obviously, there is scope to improve the methods of rearing and farming edible insects, of processing them, of increasing their shelf life, of controlling contaminants and parasites and of marketing and advertising them [23,61,62]. However, when dealing with bees, we are confident that the problems of their acceptance, be they workers or drones, is not as much of a problem as with other less familiar and therefore often despised and even feared species.

Honey bees and their products, most notably honey, but also beeswax, propolis and ‘royal jelly’, even the fermented comb refuse known as ‘mead’, are widely known and appreciated. The insect itself is one of the two domesticated species of insects and despite its painful and for some people life-threatening sting, it is not at all disliked by the public. For this reason we decided to focus on this species. We investigated the nutritional properties of the drones, because the huge numbers of them produced by the bees are not only useless to the beekeeper, but because of their association with the *Varroa* mite they are actually detrimental to the colony when around in large numbers. Even though worker bees are also nutritious [26], we do not advocate their use as human food and in fact by removing excess drones wish to boost the hive’s productivity, for it is the worker bee that carries out the bulk of pollinating our crop and fruit plants and producing the highly valued products of honey, beeswax, propolis and royal jelly.

This study of ours clearly demonstrates that the total amino and fatty acid contents are different among developmental stages. The total amount of amino acids was found to increase when prepupae turned into pupae although their weight decreases. This is in agreement with a previous report on drone bees [63] and is due almost certainly to the higher water content of the prepupae and the habit to empty their guts just prior to pupation. The development of muscle and other tissues during the pupal stage is likely to require an increase in available protein content. 

In case of *A. m. ligustica* workers significant increments in total amino acid contents had been observed to occur from larvae to pupae, but no significant differences had been found between pupae and adults although crude protein content did exhibit an increasing trend along with the developmental stage [26]. The total amino acid, i.e., protein content of the drones of both subspecies was found to be higher than that reported for different species of honey bees, e.g., *A. mellifera*, *A. cerana*, *A. dorsata* and *A. florea*. [26,27,64]. Lysine is often a limiting amino acid in cereal-based diets [65,66] and therefore the high content of lysine in drones could be advantageous with regard to the human nutritional requirement. 

Lysine is the precursor of carnitine, which plays an important role in processing fatty acids [67]. From a nutritional point of view, the three branched chain amino acids, i.e., leucine, valine and isoleucine, are important for muscle tissue maintenance while threonine is an important component of structural proteins. Histidine is the precursor of histamine, which plays a vital role in the immune response of an organism [68]. Phenylalanine and tyrosine synthesize dopamine, adrenaline (epinephrine), noradrenaline (norepinephrine) etc. (cf., [69]). Among the non-essential amino acids, glutamic acid is the precursor of γ-amino butyric acid (GABA), a neurotransmitter [70]. From the protein quality point of view, except for a few cases of the sulphur-containing amino acids (SAA) cysteine and methionine, all other essential amino acids almost satisfy the ideal protein pattern recommended by joint WHO/FAO/UNU 2007 reports [34] (Figure 2). The uncertainty involving cysteine and methionine is presumably attributable to the SAA not being entirely recovered by the acid hydrolysis process. The requirement of amino acids in the formation of tissue is higher than the requirement for maintenance of the tissues. We found that the histidine content of drones was a little less than the ideal value for tissue amino acid recommendation, although it did satisfy the maintenance recommendation.

Overall, the studied honey bee drones contained less total fatty acids, i.e., fat generally, in comparison with conventional foods of animal origin. The fatty acid contents were found within the range of previous studies on different species of *Apis* [26,27,63,64]. However, in contrast to honey bee worker pupae [26], drones belonging to the two subspecies of the present study, were found to have higher amounts of SFAs than MUFAs. This is consistent with the report on other honey bee drones [63]. The higher fat content in the prepupal and pupal stages than that of the adults is in agreement with previous observations [71,72]. In addition to trehalose and proline, fats are major substrates providing energy required for the flight of insects [73] and this may one possible reason behind the fat that the adult insect contained less fat than that of the prepupal and pupal stages. Moreover, in regards to the small amount of PUFA, not every insect is capable of synthesizing linoleic acid [74,75,76] and therefore a thorough investigation of fatty acid synthesis in the honey bee remains a further task.

High potassium and low sodium content could exhibit nutritional benefits, especially for the section of the human population suffering from hypertension [77,78]. Iron is a mineral of concern particularly in developing countries. The most vulnerable section of the population suffering from iron deficiency or anaemia are women, especially those lactating and of childbearing age. The relatively high amount of iron in drone brood, could be expected to help in ameliorating iron deficiency assuming high bioavailability. Zinc is another essential mineral that plays vital roles in many metabolic pathways including DNA replication, transcription and protein synthesis [79]. The zinc content of drone bees could be another advantage for using them and especially drone brood homogenate [80] as a human food or health ingredient. In the context of satisfying the human dietary requirement,100 g consumption of drones irrespective of developmental stage, satisfies the RDA for copper and phosphorus. It also satisfies 58.8 to 76.3% of the RDA for iron for adult male subjects, while less, namely 26.1 to 33.9% of RDA is required for females as the latter have a greater need for iron than males. In contrast, RDA for zinc is higher for males than females, and therefore, 40 to 54.5% of the RDA would be available to male subjects by consuming 100 g of drones while 55 to 75% of the RDA would be fulfilled for females (Table 4). 30.9 to 53.9% of the RDA can be attained with regard to potassium (Table 4). On the other hand, as the sodium content of drones in the present study was found to be rather small, the satisfaction level was also less. However, table salt is one of the primary sources to serve the sodium requirement of humans. A similar situation is true for calcium. Therefore, the scope for the dietary manipulation of drones and mineral fortification remains a task for further investigation.

Drones typically compose 5 to 10% of the adult population in a bee colony. The drone population is normally found at its peak in late spring or early summer. Lococq et al. [81] calculated the potential of drone brood biomass for Denmark to be approximately 80 tonnes per year and for Switzerland, it has been estimated to be about 100 tonnes per year [82]. In the beekeeping practice drones are only considered for the task of insemination of virgin queen bees and removing excess drones from the colony is an effective way to reduce or even avoid *Varroa* attacks. Apart from that, drones are treated as waste. 

Our results therefore support Ambühl [83], who in his honey bee cook book advocates drones as human food or animal feed and provides numerous recipes on how to prepare them for consumption. An interesting finding is that of Evans et al. [84], who report that in taste analyses there was a noticeable taste difference between drone larvae and pupae. A few products involving honey bees as dietary supplement are already available in Europe and elsewhere, e.g., the Romanian *Apilarnilpotent*, the Canadian *ApiDhron*^®^ and the Turkish *Harşena Apiterapi Ürünleri*. Furthermore, in Asia the Republic of Korea has recently included honey bee drone pupae in the list of edible insects [85]. Even though the protein content of the pupae is significantly higher than that of the prepupae, the biomass (weight of an individual) of the latter is higher than that of the former owing to its greater water content. A correlation between body weight and length (i.e., both higher in *A. mellifera carnica* prepupae than in *A. mellifera mellifera* prepupae) has very clearly been apparent in honey bee drones. This being so, it should explain the larger weights and sizes of *A. m. carnica* drone pupae and adult drones when compared with the corresponding developmental stages in *A. m. mellifera*. Thus, besides pupae, we propose prepupae as an economic and effective developmental stage.

## 5. Conclusions

Observing the competent nutrient composition of honey bee drone in the present study as well as a previous study [63] and noticing comparable, if not superior, nutrient content in comparisons with conventional foods of animal origin [26], we propose honey bee drone pupae and prepupae as a high quality nutritional resource. Even though the nutrient content did not vary widely between the subspecies, it did vary in connection with the developmental stages. In addition to pupae, we could propose prepupae, based on their higher biomass production, as a commercial product. As to the use of honey bee drone material as a nutritional source as well as a therapeutic agent, particularly protein is an emerging issue [80,86,87]. Thorough investigations of bioactive compounds and functionality at different developmental stages of drone brood should increase our understanding and appreciation of the potential of drone brood as a functional food; a food that can be expected to increase the acceptance of drone brood as a food item and health promoter. Previous studies, recently reviewed in [87], have abundantly demonstrated that insects are rich in nutrients, but that drone brood, similar to other insects, may be prone to microbial contamination [22,61]. Obviously this aspect needs to be given attention to Reducing the moisture content of harvested drone brood to improve its shelf-life and identifying ideal storage conditions are further aspects to be studied. However, once a standard dossier for the production of drones, maintaining hygiene and proper sanitation is available, drones can be expected to contribute to the ever-increasing demand of healthy food stuffs.

## Figures and Tables

**Figure 1 insects-12-00759-f001:**
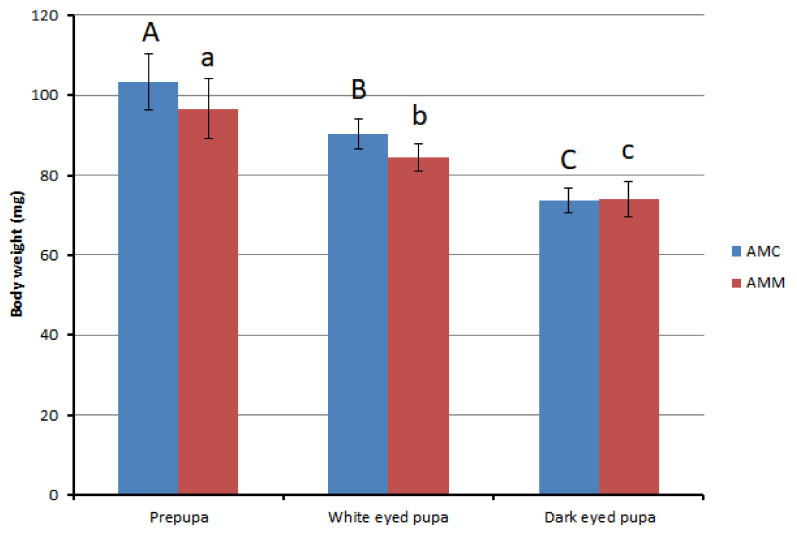
Body weights on dry matter basis (mg) of different developmental stages of AMC (*Apis mellifera carnica*) and AMM (*Apis mellifera mellifera*) drones. Different superscripts indicate statistically significant differences (*p* < 0.05) [upper case letters A, B, C indicate significant difference among prepupae, white-eyed pupae and dark-eyed pupae of *A. m. carnica*; and lower case letters a, b, c indicate the same for *A. m. mellifera*].

**Figure 2 insects-12-00759-f002:**
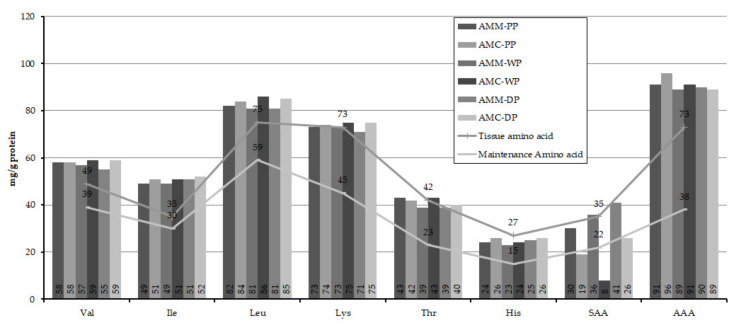
Scoring pattern of essential amino acid content of different developmental stages of *Apis mellifera carnica* and *Apis mellifera mellifera* drones taking the total amino acids as protein content. Tissue amino acid and maintenance amino acid values were obtained from the amended value of WHO/FAO/UNU 2007 report [34]. [AMM = *Apis mellifera mellifera*; AMC = *Apis mellifera carnica*; PP = Prepupae; WP = White eyed pupae; DP = Dark eyed pupae; SAA = Sulphur containing amino acids; AAA = Aromatic amino acids].

**Figure 3 insects-12-00759-f003:**
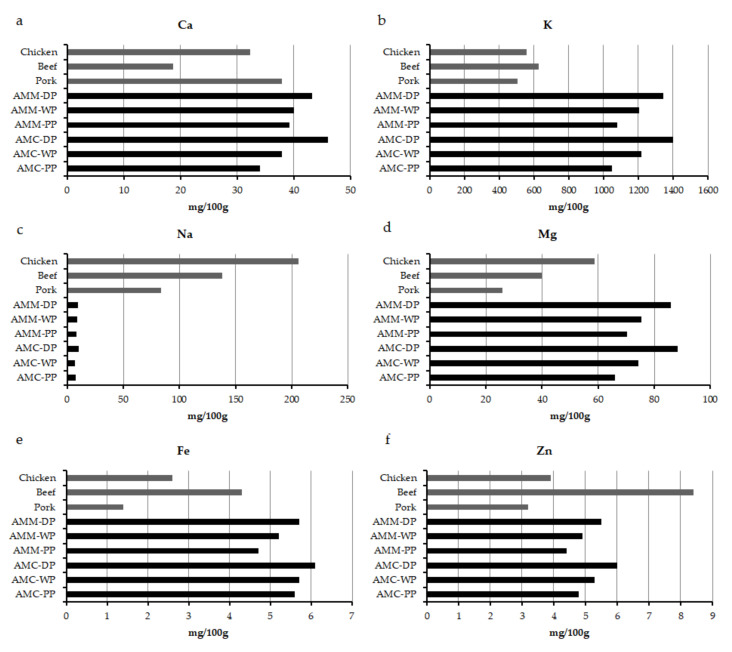
(**a**–**f**). Comparative account of selected mineral contents of different developmental stages of *Apis mellifera carnica* and *Apis mellifera mellifera* drones with conventional meats (values other than honey bee drones were obtained from USDA database [36]). [AMM = *Apis mellifera mellifera*; AMC = *Apis mellifera carnica*; PP = Prepupae; WP = White eyed pupae; DP = Dark eyed pupae].

**Table 1 insects-12-00759-t001:** Amino acid composition (g/100 g dry matter based) of different developmental stages of *Apis mellifera carnica* and *Apis mellifera mellifera* drones. One way ANOVA followed by Tukey’s HSD Post Hoc test (*p* < 0.05).

Amino Acid	*Apis mellifera carnica*					*Apis mellifera mellifera*				
	Prepupa	White Eyed Pupa	Dark Eyed Pupa	F	*p*	Prepupa	White Eyed Pupa	Dark Eyed Pupa	F	*p*
Asp	2.4 ± 0.01 ^a,A^	2.8 ± 0.02 ^b,A^	2.8 ± 0.06 ^b,A^	100.5	0.002	2.4 ± 0.00 ^a,A^	2.8 ± 0.01 ^b,A^	3.0 ± 0.00 ^c,A^	2338.0	0.000
Thr *	1.3 ± 0.03 ^a,A^	1.6 ± 0.00 ^b,A^	1.7 ± 0.02 ^c,A^	259.8	0.000	1.4 ± 0.01 ^a,A^	1.5 ± 0.01 ^b,A^	1.7 ± 0.00 ^c,A^	781.5	0.000
Ser	1.4 ± 0.02 ^a,A^	1.6 ± 0.02 ^b,A^	1.9 ± 0.01 ^c,A^	757.6	0.000	1.4 ± 0.00 ^a,A^	1.7 ± 0.02 ^b,A^	2.0 ± 0.00 ^c,A^	1024.0	0.000
Glu	6.3 ± 0.04 ^a,A^	7.7 ± 0.12 ^b,A^	7.4 ± 0.07 ^b,A^	159.1	0.001	6.6 ± 0.01 ^a,A^	7.6 ± 0.03 ^b,A^	8.1 ± 0.02 ^c,B^	2598.0	0.000
Pro	2.4 ± 0.08 ^a,A^	3.0 ± 0.03 ^b,A^	3.7 ± 0.03 ^c,A^	321.6	0.000	2.8 ± 0.01 ^a,A^	3.3 ± 0.03 ^b,B^	3.6 ± 0.00 ^c,A^	945.1	0.000
Gly	1.5 ± 0.00 ^a,A^	1.9 ± 0.02 ^b,A^	2.6 ± 0.01 ^c,A^	4588.0	0.000	1.6 ± 0.11 ^a,A^	1.9 ± 0.02 ^b,A^	2.4 ± 0.01 ^c,B^	61.4	0.000
Ala	1.5 ± 0.01 ^a,A^	2.0 ± 0.02 ^b,A^	2.9 ± 0.02 ^c,A^	2769.0	0.000	1.5 ± 0.19 ^a,A^	2.0 ± 0.01 ^b,A^	2.5 ± 0.01 ^b,B^	43.2	0.006
Val *	1.8 ± 0.02 ^a,A^	2.2 ± 0.02 ^b,A^	2.5 ± 0.01 ^c,A^	1044.0	0.000	1.9 ± 0.00 ^a,A^	2.2 ± 0.00 ^b,A^	2.4 ± 0.01 ^c,A^	4646.0	0.000
Cys	0.2 ± 0.01 ^a,A^	0.1 ± 0.01 ^b,A^	0.7 ± 0.01 ^c,A^	2063.0	0.000	0.5 ± 0.01 ^a,B^	0.6 ± 0.01 ^a,B^	1.0 ± 0.05 ^b,A^	157.2	0.001
Met *	0.4 ± 0.01 ^a,A^	0.2 ± 0.00 ^b,A^	0.4 ± 0.04 ^a,A^	61.8	0.004	0.5 ± 0.03 ^a,A^	0.8 ± 0.02 ^b,B^	0.8 ± 0.02 ^b,B^	89.5	0.002
Ile *	1.6 ± 0.02 ^a,A^	1.9 ± 0.02 ^b,A^	2.2 ± 0.04 ^c,A^	201.8	0.001	1.6 ± 0.00 ^a,A^	1.9 ± 0.04 ^b,A^	2.2 ± 0.01 ^c,A^	218.6	0.001
Leu *	2.6 ± 0.01 ^a,A^	3.2 ± 0.03 ^b,A^	3.6 ± 0.04 ^c,A^	551.9	0.000	2.7 ± 0.02 ^a,A^	3.1 ± 0.01 ^b,A^	3.5 ± 0.02 ^c,A^	1503.0	0.000
Tyr	1.6 ± 0.01 ^a,A^	1.7 ± 0.03 ^b,A^	2.0 ± 0.02 ^c,A^	256.7	0.000	1.5 ± 0.01 ^a,B^	1.8 ± 0.00 ^b,A^	2.1 ± 0.04 ^c,A^	335.8	0.000
Phe *	1.4 ± 0.00 ^a,A^	1.7 ± 0.01 ^b,A^	1.8 ± 0.00 ^c,A^	2392.0	0.000	1.5 ± 0.02 ^a,A^	1.6 ± 0.02 ^b,A^	1.8 ± 0.01 ^c,A^	227.3	0.001
His *	0.8 ± 0.06 ^a,A^	0.9 ± 0.01 ^b,A^	1.1 ± 0.00 ^b,A^	28.6	0.011	0.8 ± 0.00 ^a,A^	0.9 ± 0.01 ^a,A^	1.1 ± 0.06 ^c,A^	30.0	0.010
Lys *	2.3 ± 0.05 ^a,A^	2.8 ± 0.04 ^b,A^	3.2 ± 0.03 ^c,A^	218.7	0.001	2.4 ± 0.03 ^a,A^	2.8 ± 0.02 ^b,A^	3.1 ± 0.00 ^c,A^	958.6	0.000
Arg	1.7 ± 0.01 ^a,A^	2.1 ± 0.01 ^b,A^	2.3 ± 0.01 ^c,A^	1212.0	0.000	1.7 ± 0.04 ^a,A^	2.0 ± 0.01 ^b,B^	2.3 ± 0.02 ^c,A^	290.1	0.000
**Total**	**31.1 ^a,A^**	**37.4 ^b,A^**	**42.5 ^c,A^**	**448.8**	**0.000**	**32.8 ^a,A^**	**38.4 ^b,A^**	**43.4 ^c,A^**	**2817.0**	**0.000**

* Essential amino acid. Superscript lower case letters indicate the difference of the amino acid content among developmental stages of each subspecies and upper case letters indicate difference of amino acid content between same developmental stages of the two subspecies.

**Table 2 insects-12-00759-t002:** Fatty acid compositions (mg/100 g dry matter based) of different developmental stages of *Apis mellifera carnica* and *Apis mellifera mellifera* drones. One way ANOVA followed by Tukey’s HSD Post Hoc test (*p* < 0.05).

Fatty Acid	*Apis mellifera carnica*					*Apis mellifera mellifera*				
	Prepupa	White Eyed Pupa	Dark Eyed Pupa	F	*p*	Prepupa	White Eyed Pupa	Dark Eyed Pupa	F	*p*
**Saturated fatty acids**										
Capric acid (C10:0)	2.0 ± 0.10	1.9 ± 0.03	2.0 ± 0.03	1.5	0.289	ND	ND	1.8 ± 0.03	---	
Lauric acid (C12:0)	28.2 ± 0.84 ^a,A^	29.8 ± 0.27 ^b,A^	27.6 ± 0.20 ^a,A^	14.5	0.005	20.9 ± 0.21 ^a,B^	24.9 ± 0.30 ^b,B^	26.0 ± 0.12 ^c,B^	431.5	0.000
Myristic acid (C14:0)	379.3 ± 8.63 ^a,A^	355.0 ± 3.51 ^b,A^	234.7 ± 1.87 ^c,A^	597.9	0.000	341.7 ± 2.501 ^a,B^	354.0 ± 2.64 ^b,A^	284.1 ± 1.34 ^c,B^	833.8	0.000
Palmitic acid (C16:0)	4699.2 ± 94.42 ^a,A^	4640.6 ± 36.65 ^a,A^	3307.0 ± 32.03 ^b,A^	494.5	0.000	4847.7 ± 28.24 ^a,A^	4726.6 ± 24.57 ^b,B^	3803.9 ± 26.28 ^c,B^	1402.0	0.000
Heptadecanoic acid (C17:0)	4.2 ± 0.09 ^a,A^	4.1 ± 0.05 ^a,A^	4.1 ± 0.07 ^a,A^	1.5	0.300	4.3 ± 0.02 ^a,A^	4.7 ± 0.05 ^b,B^	4.5 ± 0.03 ^c,B^	89.1	0.000
Stearic acid (C18:0)	1277.9 ± 20.17 ^a,A^	1362.7 ± 10.37 ^b,A^	1207.3 ± 15.90 ^c,A^	71.0	0.000	1207.0 ± 7.19 ^a,B^	1260.0 ± 5.74 ^b,B^	1181.4 ± 11.29 ^c,A^	68.2	0.000
Arachidic acid (C20:0)	46.8 ± 0.35 ^a,A^	60.9 ± 0.84 ^b,A^	72.4 ± 1.54 ^c,A^	463.0	0.000	45.1 ± 0.59 ^a,B^	58.9 ± 0.69 ^b,B^	67.7 ± 0.82 ^c,B^	779.4	0.000
Behenic acid (C22:0)	16.0 ± 0.53 ^a,A^	20.9 ± 0.52 ^b,A^	30.3 ± 0.60 ^c,A^	515.9	0.000	16.9 ± 0.24 ^a,A^	21.2 ± 0.52 ^b,A^	27.6 ± 0.44 ^c,B^	490.0	0.000
**Subtotal**	**6453.7 ^a,A^**	**6475.8 ^a,A^**	**4885.4 ^b,A^**	**369.6**	**0.000**	**6483.5 ^a,A^**	**6450.3 ^a,A^**	**5396.9 ^b,B^**	**877.1**	**0.000**
**Monounsaturated fatty acids**										
Myristoleic acid (C14:1)	2.4 ± 0.07 ^A^	2.0 ± 0.03 ^A^	ND	3124.0	0.000	3.1 ± 0.02 ^a,B^	3.0 ± 0.02 ^b,B^	2.4 ± 0.04 ^c^	472.6	0.000
Palmitoleic acid (C16:1)	55.4 ± 1.21 ^a,A^	50.8 ± 0.60 ^b,A^	47.9 ± 0.96 ^c,A^	47.3	0.000	72.3 ± 0.21 ^a,B^	65.6 ± 0.34 ^b,B^	56.1 ± 0.41 ^c,B^	1815.0	0.000
Oleic acid (C18:1 n-9, Cis)	4701.8 ± 81.72 ^a,A^	4771.3 ± 39.85 ^a,A^	4316.3 ± 25.28 ^b,A^	60.7	0.000	4439.6 ± 21.05 ^a,B^	4578.5 ± 33.87 ^b,B^	4197.3 ± 38.45 ^c,B^	109.1	0.000
cis11-Eicosenic acid (C20:1 n-9)	7.3 ± 0.09 ^a,A^	7.6 ± 0.08 ^b,A^	9.1 ± 0.13 ^c,A^	247.4	0.000	6.6 ± 0.10 ^a,B^	7.6 ± 0.06 ^b,A^	8.5 ± 0.11 ^c,B^	312.0	0.000
**Subtotal**	**4766.9 ^a,A^**	**4831.7 ^a,A^**	**4373.3 ^b,A^**	**60.1**	**0.000**	**4521.6 ^a,B^**	**4654.7 ^b,B^**	**4264.2 ^c,B^**	**113.3**	**0.000**
**Polyunsaturated fatty acids**										
Linolelaidic acid (C18:2 n-6, trans)	10.2 ± 0.55 ^a,A^	13.3 ± 0.96 ^b,A^	17.3 ± 0.56 ^c,A^	73.1	0.000	21.3 ± 4.51 ^a,B^	22.0 ± 2.40 ^a,B^	22.2 ± 0.94 ^a,B^	0.1	0.920
Linoleic acid (C18:2 n-6, Cis)	46.6 ± 0.99 ^a,A^	49.0 ± 0.52 ^b,A^	36.3 ± 0.60 ^c,A^	255.4	0.000	31.3 ± 0.73 ^a,B^	53.2 ± 0.44 ^b,B^	56.8 ± 0.63 ^c,B^	1533.0	0.000
Linolenic acid (C18:3 n-3)	153.0 ± 3.51 ^a,A^	151.9 ± 1.63 ^a,A^	154.1 ± 2.27 ^a,A^	0.5	0.615	77.4 ± 0.37 ^a,B^	98.4 ± 1.14 ^b,B^	118.7 ± 2.74 ^c,B^	428.9	0.000
cis-11,14,17-Eicosatrienoic acid (C20:3 n-3)	ND	ND	1.8 ± 0.08	---		ND	ND	ND	---	
cis-13,16-Docosadienoic acid (C22:2)	14.9 ± 0.27 ^a,A^	18.8 ± 0.12 ^b,A^	26.2 ± 0.12 ^c,A^	2808.0	0.000	13.0 ± 0.49 ^a,B^	16.8 ± 0.48 ^b,B^	19.4 ± 0.09 ^c,B^	197.7	0.000
cis-5,8,11,14,17-Eicosapentaenoic acid (C20:5 n-3)	3.9 ± 0.60 ^a,A^	6.0 ± 1.06 ^a b,A^	7.3 ± 0.42 ^b,c,A^	16.0	0.004	6.5 ± 3.39 ^a,A^	7.4 ± 1.81 ^a,A^	6.6 ± 0.83 ^a,A^	0.1	0.879
**Subtotal**	**228.6 ^a,A^**	**239.0 ^b,A^**	**242.8 ^b,A^**	**11.7**	**0.009**	**149.4 ^a,B^**	**197.7 ^b,B^**	**223.8 ^c,B^**	**116.8**	**0.000**
**Total**	**11449.2 ^a,A^**	**11546.5 ^a,A^**	**9501.5 ^b,A^**	**204.4**	**0.000**	**11154.6 ^a,A^**	**11302.7 ^a,B^**	**9884.9 ^b,B^**	**413.9**	**0.000**

ND = Not detected. Superscript lower case letters indicate the difference of the amino acid content among developmental stages of each subspecies and upper case letters indicate difference of amino acid content between same developmental stages of two subspecies.

**Table 3 insects-12-00759-t003:** Mineral contents (mg/100 g dry matter based) of different developmental stages of *Apis mellifera carnica* and *Apis mellifera mellifera* drones. One way ANOVA followed by Tukey’s HSD Post Hoc test (*p* < 0.05).

	*Apis mellifera carnica*					*Apis mellifera mellifera*				
	Prepupa	White Eyed Pupa	Dark Eyed Pupa	F	*P*	Prepupa	White Eyed Pupa	Dark Eyed Pupa	F	*p*
Ca	34.0 ± 0.68 ^a,A^	37.9 ± 0.47 ^b,A^	46.1 ± 0.38 ^c,A^	415.9	0.000	39.3 ± 0.36 ^a,B^	40.1 ± 0.33 ^a,B^	43.3 ± 0.82 ^b,B^	45.9	0.000
Mg	65.9 ± 1.42 ^a,A^	74.3 ± 0.63 ^b,A^	88.4 ± 0.25 ^c,A^	465.4	0.000	70.2 ± 0.54 ^a,B^	75.3 ± 0.84 ^b,A^	85.8 ± 1.01 ^c,A^	279.4	0.000
Na	7.8 ± 0.59 ^a,A^	7.0 ± 0.01 ^a,A^	10.3 ± 0.42 ^b,A^	49.6	0.000	8.1 ± 0.25 ^a,A^	8.7 ± 0.05 ^b,B^	9.9 ± 0.05 ^c,A^	111.7	0.000
K	1048.9 ± 23.41 ^a,A^	1219.8 ± 7.26 ^b,A^	1401.2 ± 3.88 ^c,A^	453.7	0.000	1079.9 ± 4.64 ^a,A^	1205.2 ± 18.92 ^b,A^	1341.6 ± 12.48 ^c,B^	287.9	0.000
P	651.7 ± 14.92 ^a,A^	734.7 ± 3.39 ^b,A^	869.2 ± 4.62 ^c,A^	424.0	0.000	673.5 ± 3.21 ^a,A^	731.3 ± 4.18 ^b,A^	812.3 ± 12.30 ^c,B^	244.3	0.000
Fe	5.6 ± 0.65 ^a,A^	5.7 ± 0.07 ^a,A^	6.1 ± 0.04 ^a b,A^	1.0	0.407	4.7 ± 0.09 ^a,A^	5.2 ± 0.09 ^b,B^	5.7 ± 0.06 ^c,B^	119.8	0.000
Zn	4.8 ± 0.08 ^a,A^	5.3 ± 0.02 ^b,A^	6.0 ± 0.04 ^c,A^	436.6	0.000	4.4 ± 0.02 ^a,B^	4.9 ± 0.05 ^b,B^	5.5 ± 0.04 ^c,B^	597.1	0.000
Cu	1.6 ± 0.05 ^a,A^	1.8 ± 0.01 ^b,A^	2.0 ± 0.02 ^c,A^	111.6	0.000	1.5 ± 0.01 ^a,B^	1.6 ± 0.03 ^b,B^	1.9 ± 0.06 ^c,A^	76.9	0.000

Superscript lower case letters indicate the difference of the amino acid content among developmental stages of each subspecies and upper case letters indicate difference of amino acid content between same developmental stages of two subspecies.

**Table 4 insects-12-00759-t004:** Satisfying level of the RDA/AI (in %) by the consumption of 100 g of different developmental stages of *Apis mellifera carnica* and *Apis mellifera mellifera* drones (the RDA or AI of respective minerals were obtained from Linus Pauling Institute, Micronutrient Information Center, Oregon State University [35]).

	*Apis mellifera carnica*						*Apis mellifera mellifera*					
	Prepupa		White Eyed Pupa		Dark Eyed Pupa		Prepupa		White Eyed Pupa		Dark Eye Pupa	
	Male	Female	Male	Female	Male	Female	Male	Female	Male	Female	Male	Female
Ca	3.4	3.4	3.8	3.8	4.6	4.6	3.9	3.9	4.0	4.0	4.3	4.3
Mg	15.7	20.6	17.7	23.2	21.0	27.6	16.7	21.9	17.9	23.5	20.4	26.8
Na	0.5	0.5	0.5	0.5	0.7	0.7	0.5	0.5	0.6	0.6	0.7	0.7
K	30.9	40.3	35.9	46.9	41.2	53.9	31.8	41.5	35.4	46.4	39.5	51.6
P	93.1	93.1	105.0	105.0	124.2	124.2	96.2	96.2	104.5	104.5	116.0	116.0
Fe	70.0	31.1	71.3	31.7	76.3	33.9	58.8	26.1	65.0	28.9	71.3	31.7
Zn	43.6	60.0	48.2	66.3	54.5	75.0	40.0	55.0	44.5	61.3	50.0	68.8
Cu	177.8	177.8	200.0	200.0	222.2	222.2	166.7	166.7	177.8	177.8	211.1	211.1

## Data Availability

Not applicable.

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
