# Peer review of "Nutritional Composition of Honey Bee Drones of Two Subspecies Relative to Their Pupal Developmental Stages"

_insects, 2021, doi:10.3390/insects12080759_

Round 1

Reviewer 1 Report

The authors considered comments made by both reviewers and have rewritten the manuscript thoroughly. I'm OK with the current version of the manuscript.

Author Response

Reviewer 1:  The authors considered comments made by both reviewers and have rewritten the manuscript thoroughly. I'm OK with the current version of the manuscript.
Our response: Thank you very much for your time to examine our manuscript.

Reviewer 2 Report

No comments.

Author Response

Reviewer 2: No comments.

Our response: Thank you for your time to review the manuscript.

Reviewer 3 Report

The authors present a comprehensive overview of the nutritional suitability of honey bee drone prepupae and pupae for human consumption. The figures and tables are nicely organized and easy to follow and the methodology and analysis appears sound. 

L101 The sample size appears to be 20 for each developmental stage for each subspecies. Can you confirm/include in the text of the materials and methods that this was the sample size used in all subsequent analyses (sections 2.2-2.5)

L166 You indicate that there is no significant difference in body weight in the text, yet in Figure 1, it appears that there is a aignificant difference in the dark eyed pupa body weight based on the different 'C' superscripts. 

Author Response

Reviewer 3: The authors present a comprehensive overview of the nutritional suitability of honey bee drone prepupae and pupae for human consumption. The figures and tables are nicely organized and easy to follow and the methodology and analysis appears sound. 

L101 The sample size appears to be 20 for each developmental stage for each subspecies. Can you confirm/include in the text of the materials and methods that this was the sample size used in all subsequent analyses (sections 2.2-2.5)

Our response: Yes, we understand. We have mentioned that sample size at 2.1 and mentioned that this is for further chemical analyses.

L166 You indicate that there is no significant difference in body weight in the text, yet in Figure 1, it appears that there is a aignificant difference in the dark eyed pupa body weight based on the different 'C' superscripts. 

Our response: Thank you for the comment. To avoid the confusion about the superscript mentioned in the Figure 1 for statistical analysis, we have included “[upper case alphabet A,B,C indicates significant difference among prepupae, white-eyed pupae and dark-eyed pupae of A.m.carnica; and lower case alphabets a,b,c indicates the same for A. m. mellifera]”. We mentioned ‘However, no significant difference was found in connection with dried body weights in dark eyed pupae of the two subspecies’ in the text, for others there is difference. 

Also, as suggested by the Reviewer we have revised the English and corrected typographical errors which are marked by track changes. We believe that now English is perfect. In the context we would like to mention that one of the authors is a New Zealander and native English speaker, another author is from a part of India where English is an official language and a third co-author has spent many years in the USA and we have thoroughly check the manuscript.

This manuscript is a resubmission of an earlier submission. The following is a list of the peer review reports and author responses from that submission.

Round 1

Reviewer 1 Report

Research paper (Article) „Nutritional composition of honey bee drones of two subspecies relative to the pupal developmental stages” written by Sampat Ghosh , Pascal Herren , Victor Benno Meyer-Rochow , and Chuleui Jung.

The Authors have conducted interesting and elegantly simple study considering concentrations of various nutrients in the bodies of honeybee drone pupae. The manuscript has a scientific value and contains useful information. The framework, aim, and hypotheses are well reasoned, the methods are substantively correct as well as clearly and thoroughly described. However, discussion needs to be rewritten, since in the current version it is too general and vague. The Authors should undertake the significant effort to dig into available literature on concentrations of nutrients in insect bodies, and on the current state of knowledge about insects as food. Based on this the Authors should provide to the Readers deep and meaningful discussion of the results obtained in the current study via comparing these results with available literature. Below I provide more detailed comments.

Discussion is short and deficient of the relevant information. The Authors should first provide broader context for the discussion of their results: (1) what do we know about concentrations of various nutrients in insects, (2) what are current worldwide trends regarding exploitation of insects as food and are there any regional differences? (3) what are specific needs for nutrients (in numbers, e.g. daily intake, etc.) for humans in various ages and how honeybee drones could meet these needs? Further, the Authors could propose possible ways of utilization bee bodies as food, discussing caveats, advantages, disadvantages, cultural differences, etc. In Conclusions the Authors should briefly summarize most important advantages and major obstacles to the development of the field of bees and insects in general as human food. Avenues for future research as well as for application of insect food, could be proposed. Below I present brief list of most important recent literature on the topic, that hopefully will help the Authors to rewrite the discussion.

Concentrations of various nutrients in different insects:
(1) https://doi.org/10.3390/foods10051036
(2) https://doi.org/10.1016/j.ifset.2017.03.007
(3) https://doi.org/10.3390/nu13041207
(4)  https://onlinelibrary.wiley.com/doi/full/10.1111/phen.12168 (see also supporting information for this study)
(5) https://doi.org/10.3389/fevo.2016.00138 (see supplementary tables 3 and 4 for this study)
(6) https://www.ajol.info/index.php/ajb/article/view/137791 

Concentrations of nutrients in bee bodies:
(1) https://doi.org/10.1371/journal.pone.0183236   
(2) https://doi.org/10.1038/s41598-020-79647-7 (see supplementary information 2 for this study)

Journal of Insect Science has published this special collection of articles on insects as food and feed: https://academic.oup.com/jinsectscience/pages/insects-as-food-and-feed 

Review on the quality aspects of insects as food: https://doi.org/10.3390/foods8030095 

Opinion on the factors influencing consumer perception and acceptability of insect-based foods: https://doi.org/10.1016/j.cofs.2021.01.007 

Review on entomophagy: https://doi.org/10.1016/j.foodres.2019.108672 

Perceptions of eating insects among new consumer groups:
(1) https://doi.org/10.1016/j.ijgfs.2020.100268
(2) https://doi.org/10.3390/foods10040709 

Review on the future of insects as food: https://doi.org/10.3233/MNM-190348 

Review on insects as a source of alternative protein: https://doi.org/10.21323/2414-438X-2021-6-1-23-32 

Lines 205-208 – total amounts were not measured in this study. Measured were concentrations = relative amounts. Please clarify these sentences.

Lines 270-271 – prepupa has higher mass but at the same has different concentrations of some nutrients than pupa. Please elaborate on this.

Lines 275-282 – this is irrelevant considering the topic of this study. Please remove.

Reviewer 2 Report

In this study, the authors aimed to analyze the nutritional composition of two subspecies honey bee drones at developmental stages. To this end, the authors exposed drone honey bees can be utilized as human food or animal feed.

I think the significance of this article is more inclined to the biological study of the nutritional requirement of the two bee species during their development. In the introduction, very little is said about the related research progress, and more background should be given. Minor comments:

-Line 36: I missed the keywords.

-Line 128-135: Which indicators and which analysis methods are used should be described clearly.

-Line 187-191, 192-201: The format of tables 1 to 3 were incorrect. Furthermore, i missed the statistical analysis results of between same developmental stages of two subspecies.